# MoM: Mixtures of Scenario-Aware Document Memories for Retrieval-Augmented Generation Systems

## Abstract

The traditional retrieval-augmented generation (RAG) paradigm, which typically engages in the comprehension of relevant text chunks in response to received queries, inherently restricts both the depth of knowledge internalization and reasoning capabilities. To address this limitation, our research transforms the text processing in RAG from passive chunking to proactive understanding, defining this process as document memory extraction with the objective of simulating human cognitive processes during reading. Building upon this, we propose the **M**ixtures **o**f scenario-aware document **M**emories (MoM) framework, engineered to efficiently handle documents from multiple domains and train small language models (SLMs) to acquire the ability to proactively explore and construct document memories. The MoM initially instructs large language models (LLMs) to simulate domain experts in generating document logical outlines, thereby directing structured chunking and core content extraction. It employs a multi-path sampling and multi-perspective evaluation mechanism, specifically designing comprehensive metrics that represent chunk clarity and extraction completeness to select the optimal document memories. Additionally, to infuse deeper human-like reading abilities during the training of SLMs, we incorporate a reverse reasoning strategy, which deduces refined expert thinking paths from high-quality outcomes. Finally, leveraging diverse forms of content generated by MoM, we develop a three-layer document memory retrieval mechanism, which is grounded in our theoretical proof from the perspective of probabilistic modeling: compared to the strategy of fusing embeddings prior to retrieval, independently retrieving memories from each layer and subsequently fusing them can more effectively reduce information loss. Extensive experimental results across three distinct domains demonstrate that the MoM framework not only resolves text chunking challenges in existing RAG systems, providing LLMs with semantically complete document memories, but also paves the way for SLMs to achieve human-centric intelligent text processing.

## 1 Introduction

Retrieval-Augmented Generation (RAG), as a technical paradigm that combines information retrieval with generative models, effectively mitigates inherent limitations of large language models (LLMs), such as insufficient data freshness (He et al., 2022), hallucinations (Chen et al., 2023b; Liang et al., 2024), and the lack of domain-specific knowledge (Li et al., 2023). As the core architecture for knowledge-intensive tasks (Oche et al., 2025), its effectiveness is fundamentally influenced by the optimization boundaries of the retrieval mechanism. Research has shown that the quality of the retrieved text segments directly determines the upper limit of the performance of RAG systems (Lin et al., 2023; Qu et al., 2024). Optimal segmentation of documents into semantically complete and coherent segments not only enhances the generation accuracy of LLMs but also significantly improves the system's processing efficiency while reducing computational resource consumption (Xu et al., 2023; Su et al., 2024).

However, a profound cognitive gap still persists in current RAG practices. Most of these methods reduce document processing to a mechanistic preprocessing step (Gao et al., 2023; Lyu et al., 2024).

This passive approach of segmenting first and then understanding is contrary to the cognitive process of human experts (Dong et al., 2024; Singh et al., 2025). When studying a complex document, human experts actively construct a mental model: they first grasp the macro-level logical structure, then identify key arguments, and ultimately form a structured memory that is interconnected and hierarchical (Spens & Burgess, 2024; Tang & Kejriwal, 2024). To bridge this gap, we advocate for a shift in the text processing approach of RAG from passive text chunking to active memory extraction. This study aims to address the core issues arising from the traditional RAG paradigm: How can we enable the model to actively transform an unstructured text stream into a semantically complete and logically coherent structured knowledge, namely, document memory, in a manner similar to domain experts? And how can we efficiently imbue small language models (SLMs) with this deep understanding capability?

To achieve this goal, we propose the **M**ixtures **o**f scenario-aware document **M**emories (MoM) framework. Initially, to establish a macro-level understanding of the document, we instruct LLMs to assume the role of a domain-specific expert. LLMs conduct an in-depth analysis and generate a well-structured document outline. This outline not only serves as an index of the content but also lays the foundation for subsequent processing steps. Secondly, guided by the logical outline, we initiate multi-path memory extraction and evaluation. To ensure the quality of the final extraction, we design two unique metrics for comprehensive evaluation and automatic selection. This approach ensures the domain adaptability of the framework, enabling it to accurately grasp the core knowledge structure and key points of different professional documents. Thirdly, with the aim of transferring this advanced cognitive capability from LLMs to SLMs, we employ reverse engineering to construct a logically rigorous **C**hain reasoning **o**f **M**emory extraction (CoM), which can assist SLMs in thinking. Finally, to fully leverage the multi-level content produced by MoM, we develop a three-layer document memory retrieval algorithm consisting of the logical outline, core content, and the original text. Meanwhile, our theoretical analysis demonstrates that this strategy can more effectively avoid information loss, thereby achieving precise retrieval of the target knowledge.

We summarize contributions of this work as follows:

- We propose active memory extraction as an alternative to passive text chunking. By achieving a global understanding of domain-specific documents, we construct structured document memory. Additionally, leveraging reverse reasoning techniques, we enable SLMs to autonomously perform this complex task.

- We develop a three-layer document memory retrieval mechanism and provide theoretical proof from the perspective of probabilistic modeling. Compared to the traditional strategy of fusing information before retrieval, independently retrieving from different memory layers and then fusing the results can more effectively reduce information loss, thereby achieving more precise knowledge localization.

- To validate the effectiveness of the MoM framework, we conduct experiments and analyses on three datasets from different domains. By obtaining data through multiple channels, we construct 40K training samples and train multiple MemReaders. The results indicate that MoM can adaptively process texts with different structures and domains, generate high-quality document memory, and also demonstrate the feasibility of achieving human-centered high-quality text processing through SLMs.

## 2 RELATED WORKS

**Text Chunking in RAG.** As a critical prerequisite for RAG, text chunking profoundly influences the ultimate performance of the system. Mainstream RAG systems and development libraries (such as LlamaIndex and LangChain) commonly adopt traditional chunking strategies, which includes fixed-size chunking, recursive chunking, or segmentation based on grammatical boundaries like sentences and paragraphs (Guu et al., 2020; Lewis et al., 2020; Gao et al., 2023; Lyu et al., 2024). These methods are inherently context-independent and completely overlook the deep-seated semantic coherence and logical structure of the content. To overcome the aforementioned drawbacks, the research community has begun exploring semantic chunking approaches, such as merging text based on the similarity of sentence embedding vectors (Xiao et al., 2023) or decomposing text into atomic factual units like propositions (Chen et al., 2023a). Although these methods outperform traditional strategies in preserving local semantics, they generally follow a bottom-up construction logic. They

focus on the relationships between adjacent text units but lack a macroscopic understanding of the document's overall architecture. Consequently, even though they can generate locally coherent segments, when these segments are combined, they may still deviate from the document's overall theme or chapter logic, resulting in logically incomplete or biased knowledge chunks. Even attempts to use LLMs for iterative segmentation still essentially seek breakpoints locally, failing to fundamentally solve this issue while incurring substantial computational costs (Duarte et al., 2024).

**Memory of RAG.** To overcome the limitations of LLMs' finite context windows and endow them with capabilities for continuous learning and long-term interaction, constructing memory systems has emerged as a pivotal research direction in the development of RAG. However, through a systematic examination of existing memory frameworks, we find that the current research focus significantly leans towards managing short-term and long-term memory in dialogue scenarios. Represented by systems such as Mem0 (Chhikara et al., 2025), LangMem [1], MemoryScope (Li Yu, 2025), and MemoBase [2], considerable complexity has been developed in conversational memory management, exemplified by Mem0's four-stage memory updating and conflict detection, MemoryScope's importance scoring and temporal decay mechanisms, and graph-structured reasoning in Mem0g (Chhikara et al., 2025). In contrast, memory construction for documents remains at a relatively nascent stage, with relatively simplistic processing approaches. MemGPT (Packer et al., 2023) employs a paging mechanism to process information segment by segment, while Zep [3] constructs a document collection for vector-based retrieval, which essentially still adheres to the traditional RAG paradigm of chunking first and understanding later, rather than constructing a holistic, structured memory for documents. Even MemoRAG (Qian et al., 2025), which focuses on documents, primarily relies on pointers to navigate and associate between pre-segmented text fragments. This emphasis on conversations over documents reveals a research gap: the lack of an advanced mechanism for proactively constructing structured, semantically coherent memories for documents within the field. Our proposed scenario-aware document memory extraction framework can serve as a text processing module for these systems, facilitating the development of the entire field.

## 3 MoM Framework

### 3.1 Document Memory

The core objective of the MoM framework is to simulate the process by which domain experts deeply read and digest documents, transforming unstructured raw text $D$ into a structured, multi-level, and easily retrievable knowledge, which we refer to as document memory, denoted as $M_{doc}$. The entire framework can be viewed as a process of learning a mapping function $f_{MoM} : D \rightarrow M_{doc}$, which encompasses three key stages: memory extraction, CoM construction, and model training.

#### 3.1.1 Task Definition

Document memory is not merely a simple segmentation of the original text but rather a structured knowledge system that has undergone deep understanding, refinement, and reconstruction. Its core characteristics are domain-specificity, structural organization, and completeness. Formally, we define the document memory corresponding to a document $D$ as a triplet: $M_{doc} = \{O, C, A\}$, where:

- $O$ (Outline): Represents the macro-logical structure of the document. It is an ordered set composed of core topics, $O = \{o_1, o_2, \ldots, o_n\}$, providing a high-level view and indexing framework for document information organization.

- $C$ (Core Content): Represents core viewpoints of the document. It is a highly condensed set of knowledge points extracted from the content corresponding to each outline node $o_i$, $C = \{c_1, c_2, \ldots, c_n\}$.

- $A$ (Atomic Chunks): Represents the structured, fine-grained content segmentation of the original document $\mathcal{D}$ guided by $O$. $A = \{a_1, a_2, \ldots, a_n\}$ exhibit stronger semantic cohesion compared to traditional segmentation methods.

---

[1] https://github.com/langchain-ai/langmem

[2] https://github.com/memodb-io/memobase

[3] https://github.com/getzep/zep

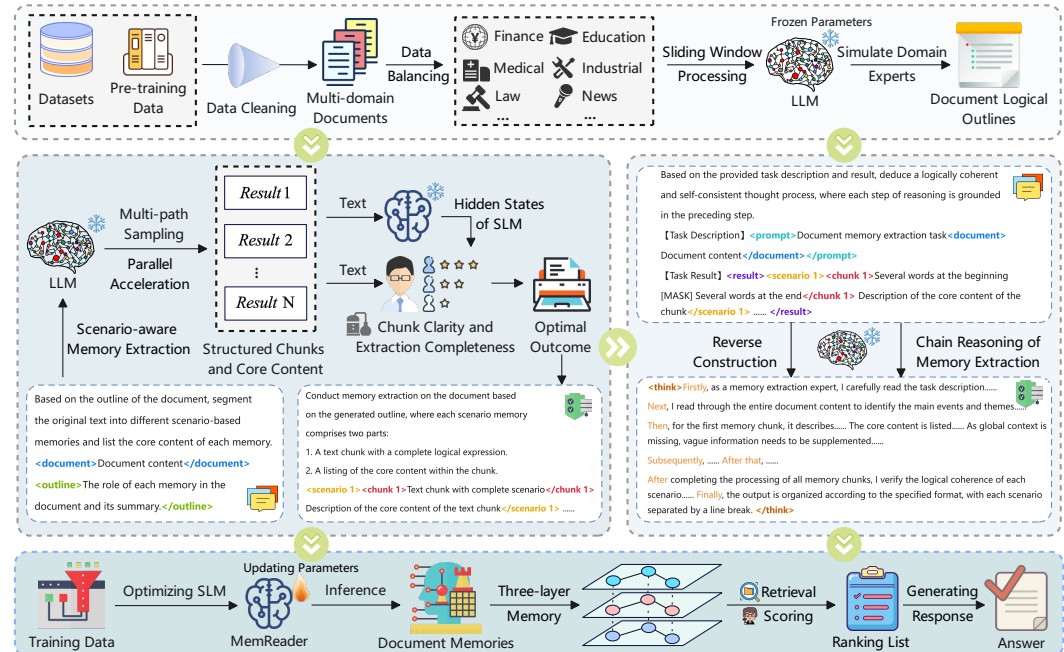

Figure 1: Overview of the entire process of our MoM framework: Logical outline generation, multi-path memory extraction and evaluation, reverse CoM construction, as well as three-layer retrieval mechanism.

### 3.1.2 SCENARIO-AWARE DOCUMENT MEMORY EXTRACTION

To generate high-quality document memory, we design an extraction process that incorporates expert simulation, multi-path sampling, and multi-dimensional evaluation.

We initially leverage a highly capable LLM, designated as the guiding model $\mathcal{M}_G$, to emulate the expertise of specialists within particular domains. Through the crafting of scenario-aware prompts, we steer $\mathcal{M}_G$ to perform a comprehensive analysis of the input document $\mathcal{D}$, resulting in the generation of its logical outline $O$. Following this, with $O$ serving as a structural framework, we instruct $\mathcal{M}_G$ to further distill and refine the core content $C$ as well as the corresponding atomic chunks $A$ for each individual outline.

Considering the randomness and limitations of single generation, we introduce a multi-path sampling strategy. By adjusting the decoding parameters of $\mathcal{M}_G$, we generate $N$ candidate document memory sets $\{M_{\text{doc}}^{(1)}, M_{\text{doc}}^{(2)}, \ldots, M_{\text{doc}}^{(N)}\}$ for the same document $\mathcal{D}$. To select the optimal solution from these candidates, we design two key quantitative evaluation metrics:

**Atomic Chunks Clarity.** This metric aims to evaluate the semantic rationality of the segmentation among atomic chunks $A$. An ideal segmentation should ensure high semantic cohesion within each chunk and clear semantic boundaries between chunks. We leverage a language model $\mathcal{M}_{\text{eval}}$ to assess the marginal probability of the semantic boundary existing between any two consecutive chunks $a_i$ and $a_{i+1}$. The clarity score is defined as follows:

$$\mathcal{S}_{\text{clarity}}(M_{\text{doc}}) = \frac{1}{n-1} \sum_{i=1}^{n-1} P_{\mathcal{M}_{\text{eval}}}(b_{i,i+1} | a_i, a_{i+1})$$

where $n$ is the total number of atomic chunks, and $b_{i,i+1}$ denotes the event that a semantic boundary exists between chunks $a_i$ and $a_{i+1}$. A higher score indicates a clearer and more logical overall chunking structure of the document.

**Core Content Completeness.** This metric is used to evaluate the effectiveness and conciseness of the core content $C$ in covering the information of the original document $\mathcal{D}$. It is achieved by

calculating the conditional perplexity of generating the entire chunks $\mathcal{A}$ given $C$, and introducing a penalty term for the length of the core content. Its formal definition is as follows:

$$\mathcal{S}_{\text{comp}}(\text{M}_{\text{doc}}) = \frac{1}{n} \sum_{i=1}^{n} \frac{1}{PPL(a_i|c_i) \cdot \log(|c_i|)}$$

where $|c_i|$ is the total number of tokens in the core content. A higher score signifies that, on average, each core content provides strong, concise support for its respective original chunk.

**Optimal Document Memory Selection.** We rank all $N$ candidate memories $\{\text{M}_{\text{doc}}^{(i)}\}$ in descending order based on $\mathcal{S}_{\text{clarity}}$ and $\mathcal{S}_{\text{comp}}$, respectively, resulting in two independent ranking lists. Subsequently, we draw on the reciprocal rank fusion algorithm (Cormack et al., 2009) to calculate a comprehensive score for each candidate. For any candidate $\text{M}_{\text{doc}}^{(i)}$, its comprehensive ranking score $\mathcal{S}_{\text{RRF}}$ is defined as:

$$\mathcal{S}_{\text{RRF}}(\text{M}_{\text{doc}}^{(i)}) = \frac{1}{k + \text{rank}_{\text{clarity}}^{(i)}} + \frac{1}{k + \text{rank}_{\text{comp}}^{(i)}}$$

where $\text{rank}_{\text{clarity}}^{(i)}$ and $\text{rank}_{\text{comp}}^{(i)}$ are the positions of $\text{M}_{\text{doc}}^{(i)}$ in the ranking lists for clarity and completeness, respectively, and $k$ is a smoothing constant (typically set to 60). All candidates will be finally ranked according to their $\mathcal{S}_{\text{RRF}}$ scores.

### 3.1.3 REVERSE CONSTRUCTION OF COM

To enable SLMs to master such complex knowledge construction capabilities, merely providing supervised data consisting of inputs $\mathcal{D}$ and outputs $\text{M}_{\text{doc}}$ is insufficient. In order to instill deeper human-like reading abilities, we introduce the reverse construction strategy of CoM. Specifically, we once again leverage the guiding model $\mathcal{M}_G$, providing it with the original document $\mathcal{D}$ and the optimal document memory $\text{M}_{\text{doc}}$, and generating the reasoning path $\mathcal{P}$ through specific prompts. This reasoning path constitutes high-quality CoM data and becomes an essential component for training SLMs.

### 3.1.4 MEMREADER

Based on the aforementioned process, we construct approximately 40K high-quality training samples by employing DeepSeek-R1 to act as $\mathcal{M}_G$. Each sample is composed of a triplet $(\mathcal{D}, \mathcal{P}, \text{M}_{\text{doc}})$. Our objective is to transform a SLM into a MemReader, enabling it to directly generate reasoning paths and document memories from raw documents. For a training sample, let the input be $s$ (the document $\mathcal{D}$ and related instructions), and the target output sequence be $o$ (the concatenation of $\mathcal{P}$ and $\text{M}_{\text{doc}}$). The loss function is defined as:

$$\mathcal{L}_{\text{F}}(\theta) = -\frac{1}{\tau} \sum_{t=1}^{\tau} \log P(o_t|o_{<t}, s; \theta)$$

where $o_t$ represents the $t$-th token in the target sequence $o$, $o_{<t}$ denotes the prefix of the target sequence up to position $t - 1$, $s$ is the input context, $\theta$ represents the learnable parameters of the SLM, and $\tau$ denotes the length of the target output sequence $o$.

### 3.2 THREE-LAYER DOCUMENT MEMORY RETRIEVAL

Based on the document memory $\text{M}_{\text{doc}} = \{O, C, A\}$ generated within the MoM framework, we construct a three-layer document memory retrieval mechanism, corresponding respectively to the global summary $O$, the core content $C$, and the original chunks $A$. This design is not merely based on empirical evidence; rather, it is also theoretically substantiated by the underpinnings of our proposed probabilistic model.

We regard the user query $q$ as a random vector sampled from a mixed distribution. Specifically, the query space $Q$ is composed of two distinct types of queries: global queries $Q_{abs}$, which are designed to represent and seek information related to the global summary $O$, and local queries $Q_{query}$, which aim to represent and retrieve details pertaining to the core content $C$, alongside enabling access

to the original chunks $A$ for more granular information when necessary. We assume that these two types of queries follow different Gaussian distributions in the embedding space: global query $q_{abs} \sim \mathcal{N}(\mu_{abs}, \Sigma_{abs})$ and local query $q_{query} \sim \mathcal{N}(\mu_{query}, \Sigma_{query})$, where $\mu$ is the mean vector and $\Sigma$ is the covariance matrix.

**Hypothesis 1** (Semantic Divergence Hypothesis)**.** *We assume that the semantic centers of global queries and local queries are significantly separated in the embedding space. Formally, it is expressed as:*

$$\|\mu_{abs} - \mu_{query}\|_2 > 0$$

*This implies that the directions of $\mu_{abs}$ and $\mu_{query}$ are significantly different, that is, their inner-product $\mu_{abs}^T \mu_{query} = \cos(\theta) < 1$.*

This hypothesis is rooted in the fundamental duality of human information-seeking behavior and linguistic expression. The user's query intentions can typically be clearly classified into macro-level comprehension and micro-level exploration, and these two types of vectors form two distinct clusters in the high-dimensional embedding space. Therefore, for the hierarchical multi-vector (HMV), its vectors $V_{abs}$ and $V_{query}$ serve as unbiased estimates of the corresponding semantic centers $\mu_{abs}$ and $\mu_{query}$, which can be expressed as $E[V_{abs}] = \mu_{abs}$ and $E[V_{query}] = \mu_{query}$. In contrast, the single-vector fusion (SVF) produces a fused vector $V_{fused}$ that constitutes a biased, compromise estimate, thereby diluting the representational purity of either semantic mode:

$$E[V_{fused}] = \mu_{fused} = (1-w)\mu_{abs} + w\mu_{query}, w \in (0,1)$$

**Theorem 1.** *For user queries, the HMV outperforms the SVF in terms of expected similarity. Specifically, we need to prove the following two points:*

- *For a global query $q_{abs}$, we have $E[q_{abs}^T V_{abs}] > E[q_{abs}^T V_{fused}]$.*

- *For a local query $q_{query}$, we have $E[q_{query}^T V_{query}] > E[q_{query}^T V_{fused}]$.*

*Since we do not need to consider the vector length and only involve the direction representation, all vectors are normalized to unit vectors for subsequent calculations.*

*Proof.* First, we analyze the expected similarity of the HMV method. According to the law of total expectation and the linearity of expectation, we can obtain:

$$E[q_{\text{abs}}^T V_{\text{abs}}] = E[E[q_{\text{abs}}^T V_{\text{abs}}|q_{\text{abs}}]] = E[q_{\text{abs}}^T E[V_{\text{abs}}]] = E[q_{\text{abs}}^T \mu_{\text{abs}}]$$
$$= E[q_{\text{abs}}]^T \mu_{\text{abs}} = \mu_{\text{abs}}^T \mu_{\text{abs}} = 1$$

Next, we analyze the expected similarity of the SVF method:

$$E[q_{abs}^T V_{fused}] = E[q_{abs}^T E[V_{fused}]] = E[q_{abs}^T \mu_{fused}] = E[q_{abs}^T((1-w)\mu_{abs} + w\mu_{query})]$$
$$= (1-w)E[q_{abs}^T \mu_{abs}] + wE[q_{abs}^T \mu_{query}] = (1-w)(\mu_{abs}^T \mu_{abs}) + w(\mu_{abs}^T \mu_{query})$$
$$= (1-w) \cdot 1 + w \cdot \cos(\theta)$$

where $\cos(\theta) = \mu_{\text{abs}}^T \mu_{\text{query}}$. According to the Hypothesis 1, $\mu_{\text{abs}}$ and $\mu_{\text{query}}$ point in different directions, so $\theta > 0$ and $\cos(\theta) < 1$. Since $w \in (0,1)$, we have $w \cdot \cos(\theta) < w$. Therefore, $(1-w) + w \cdot \cos(\theta) < (1-w) + w = 1$. That is, $E[q_{\text{abs}}^T V_{\text{abs}}] > E[q_{\text{abs}}^T V_{\text{fused}}]$. The second point can be proved in the same way. $\qquad\square$

Further, we not only demonstrate that, on average, the representation method of the HMV outperforms the SVF but also offer a more robust probabilistic guarantee through the introduction of probability bounds.

**Theorem 2.** *For any small positive deviation threshold $\epsilon > 0$, the probability that the deviation of the retrieval result of the HMV strategy from the ideal case is greater than $\epsilon$ is lower than that of the SVF strategy:*

- $P(q_{abs}^T V_{abs} < 1 - \epsilon) < P(q_{abs}^T V_{fused} < 1 - \epsilon)$

$$\bullet \ P(q_{query}^T V_{query} < 1 - \epsilon) < P(q_{query}^T V_{fused} < 1 - \epsilon)$$

In summary, the HMV method not only exhibits superior performance in terms of expectation but also maintains a consistently high similarity score with extremely low probability of obtaining a low similarity score. In contrast, the SVF model has a much higher probability of achieving a similarly low similarity score. Detailed analysis and proof are presented in Appendix A.2.

**Remark 1.** *The core insights of Theorem 1 and Theorem 2 lie in the fact that semantic fusion is a costly compromise. The SVF strategy creates a semantic average by compressing globally concepts and local details into a single vector. This fused vector occupies a compromised position in the embedding space and fails to perfectly represent the original intentions of either side. As a result, when a user query explicitly targets global or local information, its similarity to this compromised vector is necessarily lower than its similarity to a dedicated vector. Our theory rigorously explains this intuition mathematically: preserving the independence of information levels is not only beneficial but also probabilistically superior as a retrieval method because it fundamentally minimizes the loss of key features due to forced information fusion. Hence, the three-layer document memory retrieval mechanism can maximize the preservation of document information at different granularities, thereby providing more accurate and comprehensive context for subsequent generation tasks.*

## 4 Experiment and Analysis

### 4.1 Experimental Setup

**Datasets and Metrics.** The experiment primarily select CRUD (Lyu et al., 2024) in the news domain, OmniEval (Wang et al., 2024b) in the financial domain, and MultiFieldQA_zh (Bai et al., 2023) across multiple domains as the evaluation benchmarks for domain document question answering. Among them, CRUD focuses on long-answer generation; OmniEval provides manually annotated data across 5 task types and 16 financial topics, enabling a comprehensive assessment of the retrieval and generation quality of RAG systems in the vertical domain; multifieldqa_zh is derived from the LongBench benchmark. The evaluation metrics uniformly adopt the BLEU series, ROUGE-L, and METEOR, which respectively measure n-gram overlap, longest common subsequence, and the matching degree of synonyms and syntactic variations.

**Baselines.** We primarily compare MoM with six representative methods spanning from rule-based to semantic-based and then to LLM-driven approaches. Original chunking [4] merely divides long texts into fixed-length chunks while disregarding sentence boundaries. The Llama_index method (Topsakal & Akinci, 2023) maintains sentence boundaries while ensuring that the number of tokens in each chunk is close to a preset threshold. Similarity chunking (Xiao et al., 2023) utilizes sentence embedding models to partition texts according to semantic similarity, effectively grouping highly relevant sentences together. LumberChunker (Duarte et al., 2024), for the first time, introduces LLMs into the segmentation process. By using prompts, it instructs the model to judge whether there is a topic shift segment by segment and dynamically outputs the optimal segmentation points. MoC (Zhao et al., 2025) adopts a framework combining a small router and meta-chunkers to balance precision and efficiency, representing a novel paradigm for parameter-efficient chunking.

**Implementation Details.** In our approach, the construction of core training data leverages the DeepSeek-R1 [5]. To stimulate the diversity of content generated by the model, we set temperature = 0.7 and top_p = 0.8. For the MemReader implementation, we select Qwen2.5-1.5B [6] and Qwen2.5-3B [7] as base models for training. During model evaluation, we primarily utilize Qwen2.5-7B [8] and Qwen2.5-14B [9]. All language models used in the experiments are of the instruction version and are loaded with float16 precision to optimize computational efficiency. To enable retrieval-based QA, we construct a vector database using Milvus and choose bge-base-zh-v1.5 [10] as the embedding

---

[4] https://github.com/run-llama/llama_index
[5] https://huggingface.co/deepseek-ai/DeepSeek-R1
[6] https://huggingface.co/Qwen/Qwen2.5-1.5B-Instruct
[7] https://huggingface.co/Qwen/Qwen2.5-3B-Instruct
[8] https://huggingface.co/Qwen/Qwen2.5-7B-Instruct
[9] https://huggingface.co/Qwen/Qwen2.5-14B-Instruct
[10] https://huggingface.co/BAAI/bge-base-zh-v1.5

Table 1: Main experimental results are presented in three domain QA datasets. B-1, B-Avg, ROU., and MET. represent BLEU-1, BLEU-AVG, ROUGE-L, and METEOR, respectively. The best result is in bold, and the second best result is underlined.

| Chunking Methods | CRUD | | | | OmniEval | | | | MultiFieldQA | | | |
|---|---|---|---|---|---|---|---|---|---|---|---|---|
| | B-1 | B-Avg | ROU. | MET. | B-1 | B-Avg | ROU. | MET. | B-1 | B-Avg | ROU. | MET. |
| Original | 0.5022 | 0.3824 | 0.5654 | 0.7324 | 0.1906 | 0.1006 | 0.2254 | 0.3904 | 0.1707 | 0.0684 | 0.2315 | 0.3650 |
| Llama_index | 0.5312 | 0.4114 | 0.5896 | 0.7449 | 0.1969 | 0.1065 | 0.2350 | 0.4040 | 0.1732 | 0.0765 | 0.2363 | 0.3726 |
| Semantic Chunking | 0.5188 | 0.3985 | 0.5823 | 0.7434 | 0.1913 | 0.0971 | 0.2240 | 0.3821 | 0.1609 | 0.0576 | 0.2191 | 0.3468 |
| LumberChunker | 0.5061 | 0.3910 | 0.5701 | 0.7399 | 0.1997 | 0.1092 | 0.2375 | 0.4085 | 0.1841 | 0.0838 | 0.2426 | 0.3809 |
| Qwen2.5-14B | 0.5329 | 0.4127 | 0.5920 | 0.7502 | **0.2048** | **0.1140** | **0.2473** | **0.4160** | **0.1883** | **0.0890** | 0.2497 | 0.3827 |
| Qwen3-14B | 0.5382 | 0.4167 | 0.5953 | 0.7531 | 0.1907 | 0.1040 | 0.2329 | 0.4080 | 0.1800 | 0.0873 | 0.2412 | 0.3759 |
| MoC MetaChunker | 0.5456 | 0.4225 | 0.6031 | 0.7546 | 0.2042 | 0.1128 | 0.2457 | 0.4141 | 0.1707 | 0.0728 | 0.2255 | 0.3512 |
| MoM MemReader-1.5B | 0.5483 | 0.4294 | 0.6084 | 0.7597 | 0.1857 | 0.0996 | 0.2297 | 0.4049 | 0.1749 | 0.0741 | 0.2381 | 0.3814 |
| MoM MemReader-3B | **0.5556** | **0.4327** | **0.6111** | **0.7603** | 0.1960 | 0.1029 | 0.2376 | 0.4130 | 0.1842 | 0.0772 | **0.2502** | **0.3896** |

model. We set top_k= 8 to retrieve the most relevant contextual information. The hardware configuration is divided according to task requirements: model training and text processing are carried out on the NVIDIA A800 80G, while evaluation is completed on the MetaX C500 64G.

## 4.2 MAIN RESULTS

To comprehensively evaluate the effectiveness of the MoM framework, we conduct extensive experiments on three QA datasets from distinct professional domains, as detailed in Table 1. The experimental results demonstrate that, on the CRUD benchmark, our proposed Mem-Reader exhibits a dominant advantage, achieving the best performance across all four evaluation metrics. It is also noteworthy that even the smaller-scale MemReader-1.5B delivers good performance on this dataset, ranking second. To further assess the MemReader's general-

Table 2: Correlation analysis of atomic chunks clarity with ROUGE-L under different LLMs.

| Metric | ROUGE-L | Atomic Chunks Clarity | | |
|---|---|---|---|---|
| Chunking Method | | Qwen2.5-3B | Qwen2.5-7B | Qwen2.5-14B |
| Original | 0.4213 | -0.0071 | 0.0085 | -0.0510 |
| Llama_index | 0.4326 | 0.2710 | 0.1919 | 0.3531 |
| Semantic Chunking | 0.4131 | 0.1945 | 0.1447 | 0.2380 |
| Qwen2.5-14B | 0.4351 | 0.3782 | 0.3651 | 0.4996 |
| Qwen2.5-72B | 0.4405 | 0.3484 | 0.3355 | 0.4908 |
| Pearson Correlation Coefficient | | 0.7044 | 0.7585 | 0.7248 |

ization capability, we analyze its performance on two more challenging datasets: OmniEval and MultiFieldQA. On the OmniEval dataset, all methods encounter difficulties, primarily due to the substantial presence of tabular information in the financial dataset, which significantly deviates from the training documents we utilized. Although MemReader does not achieve the highest score here, its performance remains among the top tier. On the MultiFieldQA dataset, MemReader-3B once again secures the best scores on two key semantic and recall metrics, ROUGE-L and METEOR. This indicates that our method holds an advantage in accurately recalling factual information and generating semantically similar answers. Based on the comprehensive experimental results, we observe that the MoM framework demonstrates superior performance in handling pure text-based QA tasks across diverse domains, proving its ability to elevate the performance ceiling of RAG systems through proactive memory extraction and retrieval.

## 4.3 EXPLORATION OF EVALUATION METRICS FOR MEMORY EXTRACTION

During the process of memory extraction, a central challenge lies in objectively evaluating the quality of the generated memory fragments. Although traditional metrics can measure information recall to a certain extent, they are often based on retrieval QA and overlook the semantic rationality of memories themselves, thus failing to provide direct scores for rapid assessment. To address this issue, we propose atomic chunks clarity and core content completeness, both of which can directly score the memory extraction results within the MoM framework. Given the complex relationship between the former and chunk quality, we conduct the further investigation. As shown in the last

row of Table 2, under three evaluation models, the correlation coefficients between our metric and ROUGE-L reach 0.7044, 0.7585, and 0.7248, respectively, all indicating strong positive correlations. Based on this, we employ Qwen2.5-7B as the base model for metric computation when constructing the document memory dataset. Additionally, we observe that the scores for the original method are notably low, even negative in some cases, suggesting that the original paragraph segmentation is often semantically ambiguous. In contrast, chunking methods processed by algorithms achieve higher positive scores, demonstrating that these methods indeed create more semantically independent units. We also conduct tests using a larger model, Qwen2.5-72B, to ensure that atomic chunks clarity can be applied to larger-scale models.

## 4.4 How Retrieved Content Supports Answers

In the evaluation framework of RAG systems, mainstream methods typically focus on the similarity between the finally generated response and the answer. However, this end-to-end evaluation fails to clearly attribute the system's performance bottleneck to either the retrieval module or the generation module. To overcome this challenge, we design an experiment to directly quantify the informational support provided by the retrieved content for the answer. Instead of examining what the system ultimately generates, we evaluate whether the retrieved context itself contains sufficient information to derive the correct answer. Given a query $q$, the RAG

Table 3: Quantitative evaluation of informational support from retrieved memories for answers.

| Metric | Informational Support | | |
| Chunking Method | Qwen2.5-7B | Qwen2.5-14B | Qwen2-7B |
| --- | --- | --- | --- |
| Original | 3.304 | 3.341 | 2.988 |
| Llama_index | 3.343 | 3.385 | 2.935 |
| Semantic Chunking | 3.580 | 3.695 | 3.024 |
| LumberChunker | 3.337 | 3.369 | 2.906 |
| Qwen2.5-14B | 3.429 | 3.498 | 2.955 |
| Qwen3-14B | 3.264 | 3.348 | 2.903 |
| MoC MetaChunker | 3.628 | 3.671 | 3.151 |
| MoM MemReader-3B | **3.149** | **3.247** | **2.795** |

system retrieves and returns a set $\mathcal{C} = \{c_1, c_2, \ldots, c_k\}$ consisting of $k$ memories. Meanwhile, the query corresponds to a reference answer $\mathcal{A} = (a_1, a_2, \ldots, a_m)$. We define the informational support score as:

$$\mathcal{S}_{\text{support}}(\mathcal{A}|\mathcal{C}) = -\frac{1}{m} \sum_{i=1}^{m} \log P(a_i|a_1, \ldots, a_{i-1}, \mathcal{C})$$

A smaller $\mathcal{S}_{\text{support}}$ value indicates a higher likelihood of the correct answer being inferred from the retrieved memories, signifying stronger support. We conduct the test on the MultiFieldQA dataset, with the results presented in Table 3. Our proposed MoM method demonstrate superior performance across all evaluation models, which proves that the memories extracted and organized by MemReader-3B can provide more information for downstream tasks.

## 5 Conclusion

This paper aims to bridge the cognitive gap in the current RAG paradigm, namely, how to transition from passive text chunking to proactive document understanding and memory construction that simulates human experts. To address this challenge, we design and implement an innovative MoM framework, which successfully elevates document processing from superficial operations to deep cognition through a holistic solution. It begins by constructing a cognitive blueprint through the generation of a logical outline that simulates domain experts, then utilizes a multi-path extraction and evaluation algorithm to ensure the completeness and accuracy of memory content, and finally employs reverse reasoning strategies CoM to impart this complex cognitive ability to SLMs. Experimental results demonstrate that SLMs empowered by MoM exhibit superior understanding and organizational capabilities when processing multi-domain documents. Meanwhile, we propose and validate a three-layer document memory retrieval mechanism based on probabilistic modeling theory. This mechanism not only makes full use of the multi-level memories generated by MoM in engineering but also theoretically proves its superiority in reducing information loss and enhancing retrieval accuracy. Therefore, the MoM framework not only provides an effective technical pathway for optimizing existing RAG systems but also opens up new avenues for exploring how to construct SLMs that are closer to human thinking patterns and possess greater autonomous cognitive abilities.

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

## A   APPENDIX

### A.1   THE USE OF LARGE LANGUAGE MODELS

To enhance the linguistic quality and readability of this paper, we employ a LLM as an auxiliary tool during the composition of sections such as the abstract and introduction. The role of the LLM is strictly limited to refining and improving the phrasing of the authors' original drafts, with the aim of enhancing clarity and fluency of expression. All academic contributions of this paper, including all viewpoints, methodologies, and conclusions, originate solely from the authors' independent research. The author conduct a final review of all content revised with the assistance of the LLM and assumes full responsibility for it.

### A.2 PROOF OF THEOREM 2

*Proof.* The random variables of our concern are the similarity scores between the global query $q_{\text{abs}}$ and $V_{\text{abs}}$ or $V_{\text{fused}}$ under the HMV and SVF method, respectively. Let $X_{\text{HMV}} = q_{\text{abs}}^T V_{\text{abs}}$ and $X_{\text{SVF}} = q_{\text{abs}}^T V_{\text{fused}}$. According to the derivation from Theorem 1, we have $E[X_{\text{HMV}}] = 1$ and $E[X_{\text{SVF}}] = (1 - w) + w \cdot \cos(\theta) = 1 - w(1 - \cos(\theta))$.

Both the query $q_{\text{abs}}$ and the vector $V_{\text{abs}}$ are modeled as Gaussian distributions, and their inner product is also a random variable following a Gaussian distribution. We define its variance as $\sigma_{\text{HMV}}^2 = \text{Var}(q_{\text{abs}}^T V_{\text{abs}})$. Similarly, $\sigma_{\text{SVF}}^2 = \text{Var}(q_{\text{abs}}^T V_{\text{fused}})$.

Our goal is to set an upper bound on the probability that the value of $X_{\text{HMV}}$ is less than $1 - \epsilon$. Applying Hoeffding's inequality, we obtain:

$$P(X_{\text{HMV}} \leq 1 - \epsilon) \leq \exp\left(-\frac{\epsilon^2}{2\sigma_{\text{HMV}}^2}\right)$$

Let $C = w(1 - \cos(\theta))$, then the expectation of SVF can be written as $E[X_{\text{SVF}}] = \mu_S = 1 - C$.

**Case 1:** Small Deviation ($0 < \epsilon \leq C$). We standardize $X_{\text{SVF}}$:

$$P(X_{\text{SVF}} < 1 - \epsilon) = P\left(\frac{X_{\text{SVF}} - \mu_S}{\sigma_{\text{SVF}}} < \frac{1 - \epsilon - \mu_S}{\sigma_{\text{SVF}}}\right)$$

Let $\Phi(\cdot)$ denote the cumulative distribution function of the standard normal distribution. Then, we obtain:

$$P(X_{\text{SVF}} < 1 - \epsilon) = \Phi\left(\frac{1 - \epsilon - \mu_S}{\sigma_{\text{SVF}}}\right) = \Phi\left(\frac{C - \epsilon}{\sigma_{\text{SVF}}}\right)$$

In this case, $C - \epsilon \geq 0$. Therefore, $P(X_{\text{SVF}} < 1 - \epsilon) \geq \Phi(0) = 0.5$. Since $\epsilon$ is a positive constant, $P(X_{\text{HMV}} \leq 1 - \epsilon)$ remains a low-probability event constrained by an exponentially decaying term. Thus, a negligible tail probability is clearly smaller than a significant probability of at least 0.5.

**Case 2:** Large Deviation ($\epsilon > C$). This case describes a more extreme requirement. We can view both events as the extent to which the random variables deviate from their respective means.

The distance between $X_{\text{HMV}}$ and its mean 1 is at least $\epsilon$:
$$P(X_{\text{HMV}} < 1 - \epsilon) = P(X_{\text{HMV}} - E[X_{\text{HMV}}] < -\epsilon)$$

The distance between $X_{\text{SVF}}$ and its mean $1 - C$ is at least $\epsilon - C$:
$$P(X_{\text{SVF}} < 1 - \epsilon) = P(X_{\text{SVF}} < \mu_S - (\epsilon - C)) = P(X_{\text{SVF}} - E[X_{\text{SVF}}] < -(\epsilon - C))$$

Since $\epsilon > C$, we know that $\epsilon > \epsilon - C > 0$. Therefore, in practical scenarios within a specialized domain, the probability of a large deviation $\epsilon$ occurring in the specialized method HMV is much smaller than the probability of a small deviation occurring in the general method SVF. On the other hand, formally, the probability upper bound of $X_{\text{HMV}}$ decays as $\exp(-\epsilon^2)$, while the probability of $X_{\text{SVF}}$ decays as $\exp(-(\epsilon - C)^2)$. Since $\epsilon^2 > (\epsilon - C)^2$, the decay rate of $X_{\text{HMV}}$ is faster, and its probability value is smaller. $\qquad\square$

### A.3 MAIN EXPERIMENTAL DETAILS

In our experiments, we employ a total of five baseline methods, with their specific configurations detailed as follows:

(a) **Rule-based Chunking Methods**
- **Original**: This method divides long texts into segments of a fixed length, such as two hundred Chinese characters or words, without considering sentence boundaries.
- **Llama_index** (Topsakal & Akinci, 2023): This method considers both sentence completeness and token counts during segmentation. It prioritizes maintaining sentence boundaries while ensuring that the number of tokens in each chunk are close to a preset threshold. We use the `SimpleNodeParser` function from `Llama_index`, adjusting the `chunk_size` parameter to control segment length.

(b) **Dynamic Chunking Methods**

- **Similarity Chunking** (Xiao et al., 2023): Utilizes pre-trained sentence embedding models to calculate the cosine similarity between sentences. By setting a similarity threshold, sentences with lower similarity are selected as segmentation points, ensuring that sentences within each chunk are highly semantically related. This method employs the `SemanticSplitterNodeParser` from `Llama_index`. The size of text chunks is controlled by adjusting the similarity threshold.
- **LumberChunker** Duarte et al. (2024): Leverages the reasoning capabilities of LLMs to predict suitable segmentation points within the text. We utilize Qwen2.5 models with 14B parameters, set to full precision.
- **MoC MetaChunker** (Zhao et al., 2025): MoC trains a lightweight chunker model to automatically learn how to partition long texts into semantically coherent chunks without relying on fixed lengths or predefined rules. Compared to traditional heuristic methods, MetaChunker demonstrates stronger cross-task generalization capabilities, particularly in downstream tasks such as RAG, serving as a representative strong baseline approach.

## A.4 COLLECTION AND REFINEMENT OF TRAINING DATA

To construct a high-quality training set for document memory extraction, we first extract raw texts from the pre-trained corpus CCI3-HQ (Wang et al., 2024a). CCI3-HQ itself comprises 500GB of high-quality web pages and book content, encompassing approximately 100B tokens, and is accompanied by quality scores. From this corpus, we select 30K documents spanning multiple domains, including news, social media, literature, academic papers, educational and scientific popularization, legal regulations, healthcare, and more. Concurrently, we construct training and test sets from the open-source CRUD dataset. Specifically, since CRUD provides evidence context snippets corresponding to each QA pair, along with the original news repository, we can retrieve the original news articles containing these context snippets through sentence matching. Taking two-hop QA as an example, CRUD provides two news snippets, namely *news1* and *news2*, which are essential for answering the *question*. We then save the matched original news articles, *matched_news1* and *matched_news2*, that contain *news1* and *news2*, respectively. Finally, from a repository of 80K original news articles, we recall 10K news articles that contain the context snippets as the initial texts for evaluation. From the remaining documents, we randomly select 10K data samples for training. After obtaining 40K multi-domain mixed documents, we employ the MoM framework for high-quality memory extraction and CoM construction, providing a reliable foundation for subsequent supervised fine-tuning of SLMs and evaluation of document memory capabilities.

Table 4: Prompt for guiding the training and inference of MemReader in the MoM framework.

**Scenario-Aware Document Memory Extraction**

This is a document memory extraction task, and you are an expert in memory extraction. Firstly, carefully analyze the content provided below and generate a logical, self-consistent, and complete reasoning process to solve this problem. Then, utilizing the holistic understanding of the document, create a memory extraction outline, and based on this outline, generate a corresponding number of scenario memories for the given document.

The generation of the memory extraction outline should be approached from the perspective of a domain expert, leveraging the global information of the original document. Each entry in the outline should represent the role of the corresponding text chunk in the scenario memory and its summary content.

When extracting memories from the document according to the generated outline, each scenario memory should consist of two parts:

1. A text chunk with complete logical expression, segmented from the document according to logical and semantic structures. Requirements: Avoid overly short text chunks and achieve a good balance between identifying content transitions and chunk length. Each output text chunk should be composed of the first and last few characters of the chunk, with the intermediate content replaced by "[MASK]".

2. A description of the core content within the corresponding text chunk.

The overall output format is as follows:
```
<think>
```
Reasoning process
```
</think>
<outline>
```
Memory retrieval outline for the document
```
</outline>
<scenario>
<chunk>
```
First few characters of text chunk 1 [MASK] Last few characters of text chunk 1
```
</chunk>
```
Description of the core content in text chunk 1
```
</scenario>
```
.......

If you understand, reply directly with the content in the specified format, using line breaks to distinguish between different scenario memories. Do not output any other explanatory content, and do not enclose your reply in quotation marks or other delimiters.

Document content:
```
<document>
```
Document Content
```
</document>
```

