# OpenReview forum: "MoM: Mixtures of Scenario-Aware Document Memories for Retrieval-Augmented Generation Systems"
_ICLR.cc/2026/Conference — ICLR 2026 Conference Withdrawn Submission_

### Official Review · Reviewer_36o7 · 2025-10-30

**Soundness:** 2
**Presentation:** 2
**Contribution:** 2
**Rating:** 2
**Confidence:** 4

**Summary:**

The paper proposes MoM (Mixtures of Scenario-Aware Document Memories), a framework for improving retrieval-augmented generation (RAG) by replacing naive text chunking with structured, “scenario-aware” document memories. The system decomposes each document into a hierarchy of representations — outline (O), core content (C), and atomic chunks (A) — guided by a large LLM (DeepSeek) that simulates an expert reader. The authors introduce two new quality metrics, Atomic Chunk Clarity and Core Content Completeness, to select optimal memory versions, and propose a hierarchical multi-vector retrieval mechanism (HMV). Experiments on CRUD, OmniEval, and MultiFieldQA datasets report improvements over baseline chunking strategies in BLEU, ROUGE-L, and METEOR scores.

**Strengths:**

- The paper tackles an important problem — inefficient documents chunking for retrieval augmented generation.
- The method shows gains in BLEU, ROUGE, and METEOR metrics, especially on the CRUD dataset. The evaluation covers multiple baselines (LlamaIndex, Semantic Chunking, MoC MetaChunker, LumberChunker), demonstrating that MoM leads to better downstream QA performance.

**Weaknesses:**

I would not recommend this paper for acceptance as it proposes an inefficient retrieval method with several unjustified design choices that gives very marginal improvements according to surface-level metrics. The theoretical justification for the method uses unrealistic assumptions and does not look convincing in its current form.

## Marginal improvements and dataset dependence.
In Table 1 the empirical gains are modest and largely limited to CRUD (and even there gains are small, e.g. 0.7603 vs 0.7597 in METEOR score). On OmniEval and MultiFieldQA, MoM usually performs worse than other baselines, e.g., worse in any metric on OmniEval and worse in BLEU on MultiFieldQA.

## Proposed retrieval might be very inefficient in comparison to dense retrievers
Do I understand correctly that you retrieve top-k memories by S_{RRF} score (line 229)? If so, then it should require much more compute than dense retrieval: as you need to forward memories through M_{eval} for clarity score (line 209) and compute perplexity for completeness score (line 221) in contrast to one dot-product for dense retrieval. How does it compare in terms of performance and compute to dense retrieval like DPR [7]?

## Too many unjustified design choices.
Several core architectural decisions are presented without sufficient motivation or ablation evidence:
- Why is it necessary to train small “MemReader” LMs instead of directly using large models (e.g., DeepSeek) to generate memories from documents?
Please provide a comparison between your small MemReader and the baseline where memories are generated directly by DeepSeek, in terms of both QA quality and resource requirements. While I understand that a smaller LM reduces inference cost, memory generation can be precomputed and thus does not affect RAG inference latency. Moreover, training an additional model introduces considerable overhead, so the trade-off needs clearer justification.

- Why introduce two separate metrics (atomic clarity and core completeness)? Could one general coherence measure suffice?

- Why perform additional generation of Chain-of-Memories (CoM) instead of reusing the initial DeepSeek-generated reasoning traces?

- Why does memory need to have three hierarchical layers and include outline, core, and atomic parts (O, C, A)? What evidence shows that all three layers are needed rather than a subset?

## Evaluation metrics are weakly informative.

BLEU and ROUGE capture surface overlap rather than semantic accuracy. METEOR is not much better. The paper should include LLM-based evaluation of factual and semantic correctness (e.g., OmniEval-style LLM judgments).
Moreover, retrieval quality is never assessed directly. Following [1], [2] and [3], the authors should report retrieval metrics such as MRR, F1, and MAP.

## Unrealistic theoretical assumptions.

### Gaussians are poor approximation.
Real embeddings from models such as BGE, E5, Contriever, OpenAI text-embedding-3-large, etc. are non-Gaussian. They have highly anisotropic distributions, heavy tails with a few dominant directions explaining most variance [4], [5]. What's more, embedding spaces learned via contrastive losses have non-Euclidean geometry (strong curvature and clustering bias) [6]. Gaussian assumptions ignore that. Even if you relaxed the Gaussian shape, the idea of "two types of queries" is a simplification: real query embeddings form hundreds of micro-clusters (product categories, question types, syntactic templates, languages, etc.), not just two.

### Global and local clusters.
Real user queries are not cleanly separable into “macro summary” and “micro detail” clusters.
Most questions mix both aspects (e.g., “Summarize the main findings of section 3, especially the methodology details”). Even when you can label global vs local intents, their embeddings don’t form well-separated Gaussians — their overlap is large. Apart from that, in embedding space, there's no canonical "center of global meaning" or "center of local meaning."
Embedding spaces don't have interpretable axes; each dimension mixes multiple semantics.
A fixed directional separation $\mu_{\mathrm{abs}}^{\top} \mu_{\mathrm{query}}<1$ implies the semantic difference is consistent across domains, which contradicts the fact that "global vs local" is context-dependent (e.g., global summary of a 3-sentence document vs a 100-page report).


### Document embedding is an unbiased estimator.
In retrieval models, embedding vectors are not statistical estimators but nonlinear transformations trained with contrastive or cosine losses.
There's no guarantee (or even definition) that $E[V]=\mu$. Even if it was the case, during fine-tuning or RAG adaptation, embeddings are often domain-shifted and nonstationary; their expectation changes with task distribution.

## Clarity and Presentation Issues.
It remains vague what hierarchical multi-vector retrieval means and which part of the document memory is actually used for retrieval and for RAG — outline (O), core (C), atomic (A), or some combination. This is critical for reproducibility. “Hierarchical multi-vector” vs “single-vector” retrieval (line 284) is introduced abruptly, without formal definition or visualization.

[1] Lyu, Yuanjie, et al. "Crud-rag: A comprehensive chinese benchmark for retrieval-augmented generation of large language models." ACM Transactions on Information Systems 43.2 (2025): 1-32.

[2] Wang, Shuting, et al. "Omnieval: An omnidirectional and automatic rag evaluation benchmark in financial domain." arXiv preprint arXiv:2412.13018 (2024).

[3] Duarte, André V., et al. "Lumberchunker: Long-form narrative document segmentation." arXiv preprint arXiv:2406.17526 (2024).

[4] Ding, Yue, et al. "On isotropy calibration of transformers." arXiv preprint arXiv:2109.13304 (2021).

[5] Li, Bohan, et al. "On the sentence embeddings from pre-trained language models." arXiv preprint arXiv:2011.05864 (2020).

[6] Chou, Jason Chuan-Chih, and Nahid Alam. "Embedding geometries of contrastive language-image pre-training." European Conference on Computer Vision. Cham: Springer Nature Switzerland, 2024.

[7] Karpukhin, Vladimir, et al. "Dense Passage Retrieval for Open-Domain Question Answering." EMNLP (1). 2020.

**Questions:**

- Once you ranked the memories, do you include top-k of them in the context of RAG? Or you include original documents? Or you include only O or C or A? Or maybe you do something else?
- Could you please give examples of query q, corresponding ground truth document d, and O, C, and A constructed for d? It would allow us to better understand the structure of memories and how retrieval is supposed to operate.
- Which model is used to compute perplexity in line 221?
- The model $M_{\text {eval }}$ used for computing clarity (line 212) is undefined in the main text.
- Qwen 2.5 and Qwen 3 baselines are never properly explained - it is unclear whether they are LLM-based chunkers, retrievers, or both.

---

### Official Review · Reviewer_t4ew · 2025-11-02

**Soundness:** 2
**Presentation:** 2
**Contribution:** 2
**Rating:** 4
**Confidence:** 4

**Summary:**

The paper presents Mixtures of Scenario-Aware Document Memories (MoM), a framework that replaces traditional chunk-based RAG with an active memory extraction approach inspired by human reading. MoM uses LLMs to build hierarchical document memories from logical outlines, core content, and atomic chunks are evaluated through clarity and completeness metrics. It further trains smaller models (MemReaders) via reverse reasoning and employs a three-layer retrieval mechanism to reduce information loss. Experiments on multiple QA datasets show consistent improvements over existing chunking and RAG methods, highlighting MoM’s potential for more structured and cognitively aligned document understanding.

**Strengths:**

1. The paper introduces the Mixtures of Scenario-Aware Document Memories (MoM) framework, shifting RAG from passive chunking to active memory extraction, closely resembling human expert cognition.
2. The multi-stage approach has logical outlining, multi-path sampling with new evaluation metrics, reverse CoM reasoning, and three-layer retrieval and is methodically structured and theoretically supported.
3. The authors validate MoM across multiple domains (news, finance, multi-domain QA) and demonstrate consistent improvements over rule-based, semantic, and LLM-driven baselines.

**Weaknesses:**

1. The proposed clarity and completeness metrics rely heavily on large language models (LLMs) for evaluation, introducing potential bias and limiting generalization across model families.
2. The paper lacks ablation on compute cost, scalability, and robustness to noisy or structured data.

**Questions:**

1. How does the framework perform when the evaluator LLM used for clarity and completeness differs from the generator model (e.g., cross-family robustness)?
2. Can you report computational costs and latency for building and querying document memories compared to standard RAG systems?
3. How well does MoM handle non-textual or semi-structured documents such as tables, code, or scientific papers with equations?

---

### Official Review · Reviewer_yXD9 · 2025-11-03

**Soundness:** 2
**Presentation:** 2
**Contribution:** 2
**Rating:** 4
**Confidence:** 4

**Summary:**

This paper addresses the "cognitive gap" in traditional Retrieval-Augmented Generation (RAG) systems, arguing that their reliance on passive text chunking limits deep understanding and reasoning. The authors propose the MoM (Mixtures of Scenario-aware document Memories) framework, which reframes document processing as a proactive, human-like memory extraction task. The core of this framework involves using a LLM to simulate a domain expert, first generating a high-level logical outline of a document. Guided by this outline, the system then employs a multi-path sampling strategy to extract structured memories, consisting of core content summaries and semantically coherent atomic chunks. The optimal memory representation is selected using novel evaluation metrics designed to measure chunk clarity and extraction completeness. The contributions lie in both its methodology for training smaller models and its theoretically-grounded retrieval mechanism. To make this sophisticated memory extraction process efficient, the authors introduce a reverse reasoning strategy to generate "Chain of Memory" (CoM) traces, which are then used to train small language models (called MemReaders) to perform the task autonomously. Furthermore, the framework proposes a three-layer retrieval system that independently queries the generated outline, core content, and atomic chunks. The authors provide a probabilistic proof demonstrating that this hierarchical, multi-vector retrieval approach is superior to fusing embeddings before retrieval, as it minimizes information loss. Extensive experiments across three distinct domains validate the framework, showing that the trained MemReader models can effectively generate high-quality document memories and improve the overall performance of RAG systems.

**Strengths:**

[S1] This paper is well written and easy to follow.

[S2] This paper addresses important research questions (how to improve the document retrieval in the RAG context), which is potentially impactful in practical use cases.

[S3] The proposed method (MemReader) works better among the baselines (Llama_index, Semantic Chunking, LumberChunker, Qwen2.5-14B, Qwen3-14B, etc) in various aspects.

**Weaknesses:**

[W1] This paper did not use DeepSeek-R1 as a baseline (instead the authors used Qwen-2.5/3), while MoM MemReader uses DeepSeek-R1.

[W2] MoM MemReader retrieves the documents three times (for O,C,A). This can be unfair comparison to the other baselines. I.e., if other baselines are allowed three queries, they may perform much better.

[W3] MoM MemReader finetunes the base LLMs while other baselines did not. The better performance than baselines might be a bit obvious. Also, there is no comparison to commercial LLMs GPT/Gemini/Claude.

[W4] Lee et al., (https://arxiv.org/abs/2402.09727) propose ReadAgent, which is a relevant baseline and the paper missing to compare. ReadAgent summarizes the long document by proactively chucking and shows better performance than RAG in long context tasks.

[W5] While automated metrics, such as ROUGE/BLEU have been used for a while, it may not always quantify the good results in NLP tasks. It would be better to include the evaluation with "LLM as a judge" to reflect the qualitative comparison aspects more.

**Questions:**

[Q1] How the retrieved documents and the final answers are qualitatively different among the baselines and proposed methods. I wonder how the difference in the evaluation metrics correlates with the actual performance.

[Q2] Which dataset is used for the training of MoM MemReader? If I missed it, please point out the line number.

[Q3] Could you include the citation for ROUGE score and METEOR score?

---

### Official Review · Reviewer_y9Z4 · 2025-11-03

**Soundness:** 3
**Presentation:** 3
**Contribution:** 3
**Rating:** 4
**Confidence:** 3

**Summary:**

The paper proposes a hierarchical representation for documents in a retrieval corpus and training method to create light-weight LMs capable of creating this hierarchical representation. In particular, each document in the retrieval corpus is represented as:
1. Outline: Course document structure and high-level description of document contents
2. Core Content: For each entry in the outline, this layer maintains a short description of the key content in the chunks associated
3. Atomic Chunks: The actual chunks of the document that convey the information summarized by the core content level

The paper trains small LMs (Qwen-2.5-1.5B and Qwen-2.5-3B) based on a labeled dataset constructed using a teacher model. The teacher model (DeepSeek-R1) proposes multiple candidate chunkings of the document. Each potential chunking is scored for (1) atomic chunk clarity (whether the boundaries between atomic chunks are reasonable) and (2) core content completeness (how well can the atomic chunk information be derived from the summarized core content). The best chunking among the candidates is chosen by reciprocal rank fusion of the two scores. The teacher model is then prompted to generate a rationale for the chunking. This method is used to generate a chunking dataset of 40K samples. The small LMs are then trained to generate the rationale and chunking given the source document resulting in the "MemReader" models.

The trained MemReader model is used to chunk the corpora for 3 challenging RAG benchmarks: CRUD, OmniEval (tex-only) and MultiFieldQA. Next, the authors argue that retrieval should account for each representation separately instead of jointly representing a chunk across all three layers. The Mixture of Memories (MoM) framework is proposed to retrieve document chunks. Theoretical proofs are provided for why fused representations of chunks lose information relative to maintaining a multi-vector representation to motivate MoM.

MoM MemReader is shown to be competitive with or outperform other document chunking approaches in end-to-end QA. Additional experiments show that the Atomic Chunk Clarity training signal is well correlated with downstream QA quality. Moreover, a reader-independent metric (Information Support) is used to show that the reference answer likelihood using documents retrieved by MemReader is higher than baselines.

**Strengths:**

1. MemReader provides a framework to train small LMs using the signal from an LLM combined with well-motivated scoring functions. Building scalable chunking algorithms is necessary for widespread adoption
    - The Atomic Chunk Clarity scoring function is demonstrated to correlate with downstream QA performance
2. The reader-model independent scores (Information Support) are useful for establishing the quality of document chunks.

**Weaknesses:**

1. The implementation details of MoM are missing making it difficult to tie the theoretical proofs to the implementation
    - How are hierarchical multi-vector representations generated? Do you embed each level of the MemReader chunking output (outline, core content, atomic chunks) separately using the document embedding model?
    - How do you retrieve documents from the corpus? Do you sum the scores from each level to obtain chunk level scores?
    - What is retrieved? Can you retrieve outline / core content without retrieving atomic chunks? If so, how do you ensure that the core content is faithful to the facts in the atomic chunks?
2. Issues with the assumptions in the theory:
    - Hypothesis 1: The theory assumes that the query representations for coarse and fine-grained queries lie in separable Gaussians. This is a property of the query embedding model which is not trained in this paper. How is this a valid assumption to make for general query embedding models?
3. Benchmark scores for several systems are close. Please provide confidence intervals for Tables 1 and 3

**Questions:**

## Clarifications

1. Table 1: How are Qwen2.5-14B and Qwen3-14B used for chunking? Where is this described in the paper text?
2. Table 1: What is the reader model (generative QA model) for the results?
3. Line 368: What evaluations are the Qwen models needed for?

---

### Note · Authors · 2025-12-02

**Comment:**

We express our gratitude to the reviewers for their feedback and the time they have devoted to reviewing our work.

**Withdrawal Confirmation:**

I have read and agree with the venue's withdrawal policy on behalf of myself and my co-authors.